# Wearable in-sensor reservoir computing using optoelectronic polymers with through-space charge-transport characteristics for multi-task learning

Xiaosong Wu [1,5], Shaocong Wang [2,5], Wei Huang[1], Yu Dong [1], Zhongrui Wang [2] ✉ & Weiguo Huang [1,3,4] ✉

In-sensor multi-task learning is not only the key merit of biological visions but also a primary goal of artificial-general-intelligence. However, traditional silicon-vision-chips suffer from large time/energy overheads. Further, training conventional deep-learning models is neither scalable nor affordable on edge-devices. Here, a material-algorithm co-design is proposed to emulate human retina and the affordable learning paradigm. Relying on a bottle-brush-shaped semiconducting $p$-NDI with efficient exciton-dissociations and through-space charge-transport characteristics, a wearable transistor-based dynamic in-sensor Reservoir-Computing system manifesting excellent separability, fading memory, and echo state property on different tasks is developed. Paired with a 'readout function' on memristive organic diodes, the RC recognizes handwritten letters and numbers, and classifies diverse costumes with accuracies of 98.04%, 88.18%, and 91.76%, respectively (higher than all reported organic semiconductors). In addition to 2D images, the spatiotemporal dynamics of RC naturally extract features of event-based videos, classifying 3 types of hand gestures at an accuracy of 98.62%. Further, the computing cost is significantly lower than that of the conventional artificial-neural-networks. This work provides a promising material-algorithm co-design for affordable and highly efficient photonic neuromorphic systems.

The human retina not only senses but also simultaneously processes light signals by harvesting their rich dynamics, which speeds up the task-dependent learning in the down-stream visual cortex[1–3]. The synergy of both the retina and the visual cortex is the basis of the efficient, compact, and fast-learned multi-tasking capability of the brain[4–7], also an essential goal of artificial general intelligence (AGI)[4,8–10]. In contrast, traditional silicon-vision chips with physically separated sensing, processing, and storage units suffer from the large

[1]State Key Laboratory of Structural Chemistry, Fujian Institute of Research on the Structure of Matter, Chinese Academy of Sciences, 350002 Fuzhou, Fujian, P. R. China. [2]Department of Electrical and Electronic Engineering, University of Hong Kong, Pokfulam Road, Hong Kong SAR, P. R. China. [3]Fujian Science & Technology Innovation Laboratory for Optoelectronic Information of China, 350002 Fuzhou, Fujian, P. R. China. [4]University of Chinese Academy of Sciences, 19A Yuquan Road, 100049 Beijing, P. R. China. [5]These authors contributed equally: Xiaosong Wu, Shaocong Wang. ✉e-mail: zrwang@eee.hku.hk; whuang@fjirsm.ac.cn

time and energy overheads incurred by the massive and frequent data shuttling among these units, as well as sequential analog-digital conversions, a fundamental limit of the underlying energy-efficiency[11–18]. This is further intensified by the slowdown of Moore's law, which imposes a challenge to their footprint[11,19,20]. Moreover, the learning in conventional deep-learning models, such as recurrent neural networks for temporal signals[21–23], employ tedious training (e.g., gradient descent via backpropagation through time, BPTT) on a very specific task, that is neither scalable nor affordable on edge-devices with limited battery access and form factor.

Tremendous efforts have been dedicated to emulating the human retina and the affordable learning paradigm. Material-wise, inorganic light-responsive 2D semiconductors, such as $MoS_2$ with defects and impurity sites[15,24], SnS bearing dual-type defect states associated with Sn and S[25], layered black phosphorus-containing oxidation-related defects[1], perovskite quantum dots exhibiting strong photogating effect[26–28], $h$-BN/$WSe_2$ heterostructure capable of electron trapping and de-trapping[29], and $MoO_x$ showing valence state changes, are the most widely used materials for artificial retinas[2]. In addition, organic semiconductors, that are intrinsically bio-compatible, wearable, and scalable[13,30,31], such as PDVT-10[32,33], chlorophyll doped PDPP4T[34], and pentacene/silk&CDs bilayer have emulated the biological counterpart in a more faithful manner[35]. Algorithm-wise, reservoir computing (RC), which nonlinearly casts temporal signals to a feature space by harvesting the fading memory of a fixed dynamic system, was deemed a promising solution for affordable edge learning[25,36,37]. As the learning of a RC is limited to the readout layer of long-term memory, the training cost is significantly reduced compared with that of conventional deep-learning models[25,36]. However, it is still yet to devise a paired material-algorithm to combine the efficient artificial retina and affordable RC-based edge learning, therefore unleashing the multi-tasking potential of bio-inspired neuromorphic vision.

Here, we proposed a material-algorithm co-design, a light-responsive semiconducting polymer ($p$-NDI) with efficient exciton dissociations and through-space charge-transport characteristics to construct an in-sensor RC for multi-tasked pattern classification[38–40]. The flexible neuromorphic device is based on a three-terminal transistor with a $p$-NDI semiconductor channel. Attributing to its excellent light-responsive behavior and nonlinear fading memory, this device is able to in situ sense, memorize, and pre-process the optical inputs (i.e., contrast enhancement and noise reduction) simultaneously. Moreover, the synergies between the exciton dissociation/charge recombination dynamics, the photogating effects, and the through-space charge-transport characteristics in the polymers enable a transistor-based dynamic RC system manifesting excellent separability, fading memory, and echo state property on different tasks. These RC-based retinas are paired with a 'readout function' implemented on memristive organic ionic gel diodes. The synergistic function of signal pre-processing and dynamic RC offered by all-organic optoelectronic materials (i.e., $p$-NDI and organic ionic gel) achieves accuracies of 98.04%, 88.18%, and 91.76%, respectively (higher than all reported organic semiconductor) in recognizing handwritten letters and numbers, and classifying a variety of costumes, which translates to the multi-tasked learning of costume style and size. The overall accuracy is 88.00% for the system to correctly recognize not only the garments but also their size. Despite 2D images, the spatiotemporal dynamics of the RC is leveraged to classify the event-based videos about left-hand waving, right-hand waving, and hand clapping gestures, showing an accuracy of 98.62%. Note that this $p$-NDI transistor-based RC is free from liquid electrolytes that are widely used in synaptic organic electrochemical transistors, rendering augmented scalability and handleability[4,12,30,37]. This work provides a promising material-algorithm co-design strategy for wearable, affordable, and highly efficient photonic neuromorphic systems with multi-task learning capability.

## Results

### Semiconductor design for optoelectronic in-sensor RC with multi-task learning capability

To construct an in-sensor RC that emulates the multi-tasked optical signal process function of the human visual system, the semiconductor has to meet the following requirements: (1) providing well-separated photocurrent responses to light irradiations of different intensities and pulse widths, rather than a fast-saturated response; (2) possessing a fading memory featuring a nonlinear decay behavior after switching off the light; (3) maintaining a good electrical switching behavior upon light irradiation. However, most conventional semiconductors are incompetent in these aspects. For example, most inorganic semiconductors are unable to give well-separated responses. Organic semiconductors such as pentacene and poly(3-hexylthiophene-2,5-diyl) (P3HT) feature fast-saturated photocurrents when being irradiated and instant current decay upon light removal (Fig. 1 and Supplementary Fig. 1). Although P3HT/$SiO_2$-based phototransistor shows a slow photocurrent decay after switching off the light (Supplementary Fig. 2f), this memory behavior is stemmed from the surficial charge traps of the $SiO_2$ dielectric layer, rather than the P3HT itself[41,42]. As P3HT serves as a charge carrier transport material rather than a charge storage/trapping material. Removing these surficial charge traps by self-assembled monolayer (SAM) treatment of $SiO_2$ substrates can remarkably promote charge carrier recombination and eliminate the memory of P3HT, leading to a typical photodetector behavior (Supplementary Fig. 2d, e)[41,42]. Moreover, P3HT-based phototransistor generates a high off-current upon light irradiation (Supplementary Fig. 1c). In fact, many other conventional conjugated polymers show similar behaviors[43–45]. The very high off-currents not only yield large power consumptions and small on/off ratios, but also give rise to poor electrical switching and modulation capabilities, and an inferior stability[46–50]. All the above factors disqualify P3HT and many conventional organic materials for a high-performance in-sensor RC system (Fig. 1 and Supplementary Fig. 1). Surprisingly, $N, N'$-dioctyl-naphthalenetetracarboxylic diimide (C8-NDI) hardly responds to light irradiation, as evidenced by the negligible current change (Supplementary Fig. 1g, h). The aforementioned different photo-responsive behaviors indicate that a crystalline form of organic semiconductor (i.e., pentacene and C8-NDI, Supplementary Fig. 3) hampers exciton dissociation, and/or accelerates charge recombination, which therefore quenches the photocurrent and gives rise to a fast-saturated photocurrent or even no response upon light irradiation[51,52]. As a result, to develop a high-performance semiconductor for in-sensor RC, a delicate molecular design is required to balance the exciton dissociation rate, charge recombination, and diffusion/transport behaviors. To this end, we adopt a semiconductor configuration in which the molecular π-planes are stringed together on a non-conjugated polymer backbone with a well-defined interval distance. Theoretically, the flexible polymer backbone (1) prevents long-range ordering of molecular π-planes but leads to short-range ordered domains with different molecular orientations, which generates an electrostatic landscape with an interfacial energy offset, and facilitates exciton dissociation and charge separation[53,54]. Such a feature is critical to achieving a nearly-linear and continuous response to light irradiation, rather than a fast-saturated photocurrent response that is typically observed in conventional semiconductors (e.g., P3HT and pentacene). As a result, a well-separated photocurrent response that is essential for an in-sensor RC system could be achieved. (2) Forms a through-space charge-transport (TSCT) channel that renders the charge carriers with reasonable mobilities and fading memories[55–57]. Note that TSCT mode of $p$-NDI is intrinsically different from the bandlike and through-bond charge-transport (TBCT) modes in conventional semiconductors. The TSCT mode enables an efficient charge carrier transport at ordered domains, but a poor charge carrier transport at disordered domains which serve as charge storage/trapping sites, giving rise to a slow recombination of

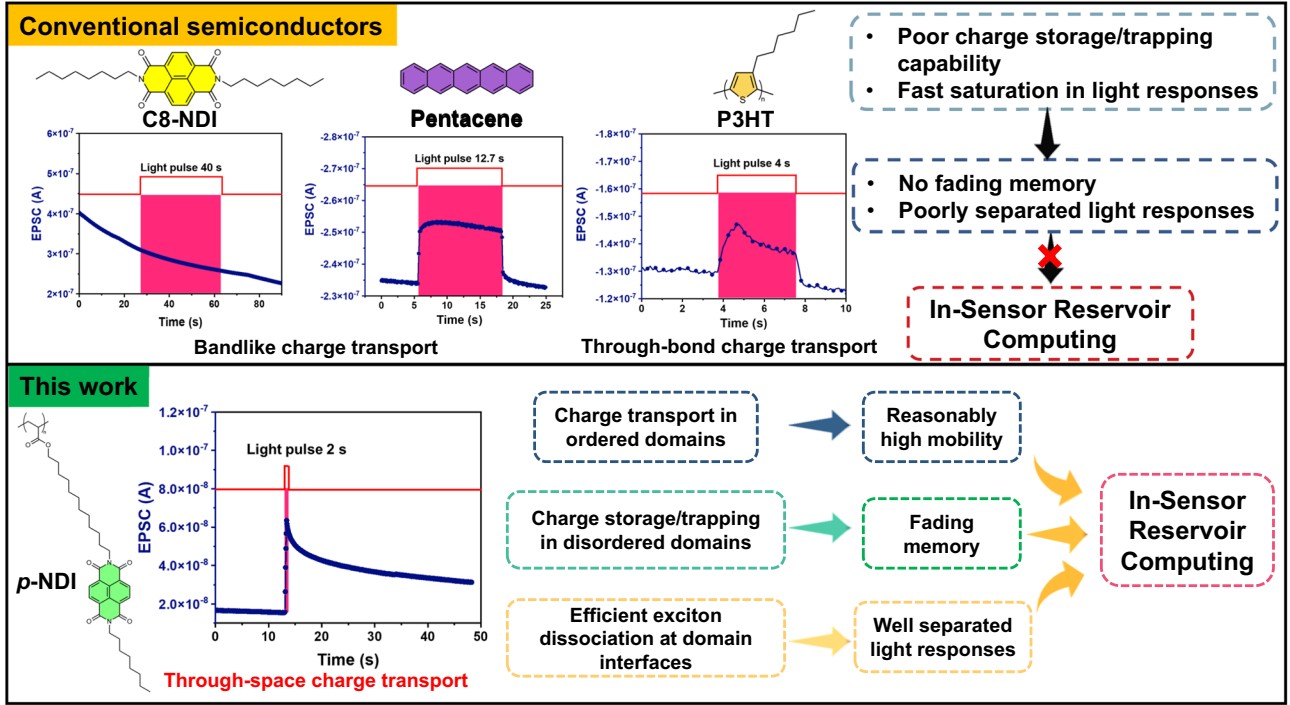

**Fig. 1 | Comparison of the photocurrent responses of conventional semiconductors and *p*-NDI, and the detailed semiconductor design principle for in-sensor RC systems.** The photocurrent responses of C8-NDI, pentacene, and P3HT, which are incapable for in-sensor reservoir computing. The photocurrent response of *p*-NDI shows a fading memory and is suitable for in-sensor reservoir computing.

charge carriers and thus a fading memory. However, the bandlike and TBCT modes in conventional semiconductors only facilitate charge carrier transport rather than charge carrier storage/trapping[58]. Due to the lack of charge carrier storage/trapping sites, P3HT and many conventional semiconductors suffer from a fast recombination of opposite charge carriers and do not show a fading memory. (3) Offers good flexibility and thermal stability to the resulting semiconductor. As shown in Fig. 1, *p*-NDI ($M_n$ ~ 28,000 Da, Supplementary Fig. 4) containing NDI units as the side groups and polyamide as the backbone is developed according to the above design principle. Its chemical structure is unambiguously confirmed by both $^1$H NMR and $^{19}$F NMR, as well as Attenuated Total Reflection (ATR) characterization (Supplementary Figs. 5, 38, and 39). Thermogravimetric Analysis (TGA) indicates a superior thermal stability of *p*-NDI with a decomposition temperature ($T_d$) over 400 °C (Supplementary Fig. 6a). Differential scanning calorimetry (DSC) measurement reveals a glass transition temperature ($T_g$) of 165 °C, a melting temperature ($T_m$) ranging from 200 to 225 °C, and a crystallization temperature ($T_c$) of 200 °C, respectively (Supplementary Fig. 6b). The UV-vis absorption spectra of *p*-NDI well-overlaps with that of C8-NDI, implying the polymer backbone impacts negligibly to the electronic structure of NDI units (Supplementary Fig. 7). The respective highest occupied molecular orbital (HOMO) and lowest unoccupied molecular orbital (LUMO) energy level of *p*-NDI is –6.01 and –2.92 eV, as measured by Ultraviolet Photoelectron Spectroscopy (UPS), manifesting an energy bandgap of 3.09 eV (Supplementary Fig. 8).

**Electrical properties and photo-responsive behaviors of *p*-NDI**
The three-terminal field-effect transistors with *p*-NDI as the semiconducting layer show a typical *n*-type switching behavior with a good reproducibility (Fig. 2a, b, and Supplementary Fig. 9). The typical electron mobility ($\mu$), threshold voltage ($V_{th}$), and on/off ratio are 0.01 cm$^2$V$^{-1}$s$^{-1}$, 22 V, and 10$^6$, respectively (Supplementary Fig. 10). Note that the mobility is among the highest reported in the literature of bottle-brush-shaped semiconductor molecules,

indicating an excellent through-space charge-transport ability. In addition, the *p*-NDI transistors exhibit much better mechanical and thermal stabilities than C8-NDI-based transistors, showing negligible noise at 37 °C and good long-term stability, which rationalizes the proposed semiconductor design strategies (Supplementary Fig. 11). Upon light irradiation, the source-drain current ($I_d$) continuously increases over the light intensities or pulse widths (Fig. 2c, d), rather than a fast saturation in the photocurrent response. Whereas the transistor with C8-NDI semiconductor does not respond to light irradiation (Supplementary Fig. 12). The photoresponsivity ($R$), detectivity ($D^*$), and photo/dark current ratio ($P$) of *p*-NDI transistor are detailed in the supporting information (Supplementary Fig. 13)[34,45,59]. After the removal of light, the $I_d$ decreases nonlinearly over time, resulting in a fading memory (Fig. 2c, d). Moreover, the *p*-NDI transistor exhibits typical synaptic behaviors (Supplementary Figs. 2, 14), e.g., spike-rate dependent plasticity (SRDP), paired-pulse facilitation (PPF), and paired-pulse depression (PPD)[34,48]. All the above behaviors are crucial for optical signal pre-processing and optoelectronic in-sensor RC construction.

Note that the photo response and decay rate of the transistors could also be readily adjusted by semiconductor engineering to better accommodate high-frequency optical pulses. As mentioned above, the TSCT characteristics in *p*-NDI semiconducting layer gives a slow recombination of charge carriers and thus a fading memory. In sharp contrast, the high charge recombination rate in the highly crystalline C8-NDI semiconducting layer leads to a fast photocurrent quenching and no fading memory. Leveraging the complementary behaviors *p*-NDI and C8-NDI, the photocurrent decay rate could be readily adjusted by blending C8-NDI and *p*-NDI of the semiconducting layer at different ratios. To this end, we fabricated a series of phototransistors with C8-NDI/*p*-NDI blend semiconducting layer at respective C8-NDI weight percentage of 2.5%, 5%, 10%, 25%, 70%, and 90%, and then characterized their photo responses and decay behaviors at identical conditions. As a control, the data of 0% C8-NDI, i.e., solo *p*-NDI semiconducting layer, is also included. As shown in Supplementary

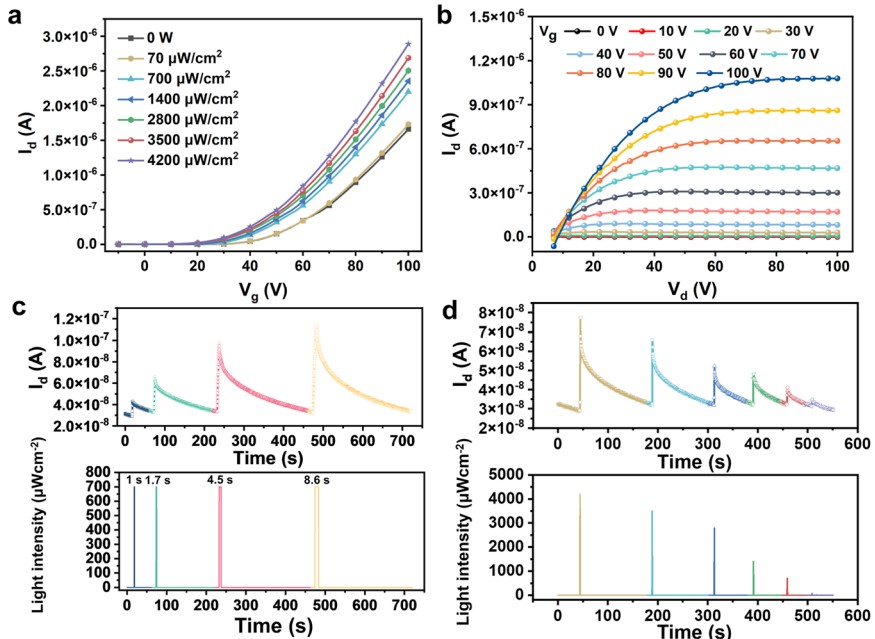

**Fig. 2 | Electrical characterizations of *p*-NDI based transistor. a** The transfer curves of *p*-NDI based transistor under different light intensities for 5 s ($V_d$ = 100 V). **b** The typical output curve of *p*-NDI based transistor under dark. The photocurrent response of *p*-NDI based transistor **c** under 700 μW cm⁻² light irradiation with different pulse widths, **d** under 1 s light irradiation with different intensities. In panels **c** and **d**, the transistor was working in a 'sampling' mode at a constant $V_g$ and $V_d$ of 100 V with a holding time and interval of 0.01 s and 0.1 s, respectively.

Fig. 15, the time required to fully recover the $I_d$ to the baseline decreases notably from over 55 s to 37, 26, 17, 8, 5, and 3 s when increases C8-NDI weight percentage from 0% to 2.5%, 5%, 10%, 25%, 70%, and 90%, respectively, implying a robust tunability of the photocurrent decay rate. Such tunability is beneficial for processing optical pulses with higher frequencies.

Apart from the semiconductor engineering strategy, device resetting could also be a way to clear the residual photocurrent of the previous light pulse. As shown in Supplementary Fig. 15, the slowly decaying photocurrent after light removal undergoes a rapid recovery upon receiving an electrical pulse ($V_g$ = –100 V, 0.1 s). The device after reset is ready for receiving the next optical pulse train and can repeatedly respond to new optical pulses with similar initial photocurrents. Moreover, the timing and frequency to reset can be customized according to the underlying optical pulse frequencies, application scenarios, and reservoir parameters.

**Mechanism study for the photo responses and fading memory of *p*-NDI**

To gain deep insights into the mechanism of the unique photo responses and fading memory of *p*-NDI, we first conduct a morphological study of the *p*-NDI films. Both Scanning Electron Microscopy (SEM) and Atomic Force Microscopy (AFM) characterizations reveal a rather amorphous feature of *p*-NDI films (Fig. 3b, d), as evidenced by the nearly featureless morphologies. In sharp contrast, C8-NDI forms well-orientated and tightly packed large crystal grains, as expected for a highly planar molecule with a large π conjugated system (Fig. 3a, c). This morphological difference evidences that the polymer backbone of *p*-NDI hampers the long-range ordering of NDI π-planes. Further information on the ordering of *p*-NDI films was obtained using 2D grazing-incidence X-ray diffraction (GIXD). In the diffraction pattern, the presence of broad arcs for diffraction spots along $q_{xy}$ corresponding to different planes implies a high degree of in-plane disordering. The three peaks with "*d*" of 6.21, 4.74, and 3.42 Å could be assigned as 010, 100, and π-π stacking, respectively, according to the diffraction pattern of C8-NDI-C12 films (Supplementary Fig. 16)[60].

The in-plane π-π stacking is of great importance for an efficient charge carrier transport for the transistor[61]. Quantitative determining the in-plane 2D order parameter ($S_2$) of *p*-NDI film is conducted as well. The azimuthal intensity variations of 100 peak are used to determine $S_2$ using the following equation:

$$S_2 = 2\langle\cos^2(\varphi)\rangle - 1 \qquad (1)$$

Where $\varphi$ is the misorientation angle of a given molecule from the average alignment direction[62]. We obtained a $S_2$ of 0.314 (as described in more detail in Supplementary Fig. 17a), manifesting a low degree of in-plane ordering[62]. This result is consistent with the Transmission Electron Microscopy (TEM) characterization of *p*-NDI film, which shows densely packed random polymer coils with different orientations (Fig. 3i). Such a high degree of disordering generates an electrostatic landscape with an interfacial energy offset at the boundaries between *p*-NDI coils, which promotes the formation of hybridized exciton/charge-transfer states at the interface, resulting in an efficient exciton dissociation into free charges[53,54]. In addition to the diffraction arcs along $q_{xy}$ direction, the presence of two bright short arcs along $q_z$ direction reveals a high degree of out-of-plane ordering. Similarly, a $S_2$ of 0.70 is obtained by analyzing the azimuthal intensity profiles from the GIXD pattern of these two diffraction peaks (Supplementary Fig. 17b)[62]. In particular, the respective "*d*" of 31.60 and 20.03 Å of the two short diffraction arcs also indicate the polymer backbones lay parallelly on the substrate, whereas the C8-NDI-C12 side chains lay obliquely related to the substrate with an inclined angle of 46° (Fig. 3j)[60]. Further, the neighboring C8-NDI-C12 side chains franking the polymer backbone are stretched in opposite directions. Along the out-of-plane direction, *p*-NDI molecules are directly piled up on top of the other without side chain interdigitations (Fig. 3f). The high degree out-of-plane orientation creates a well-connected pathway for an efficient through-space charge transport[55–57]. During the photoexcitation process, charges are generated throughout the *p*-NDI film. The majority of mobile carriers are electrons which contributes to the rise of photocurrent, while the holes are localized immediately after being

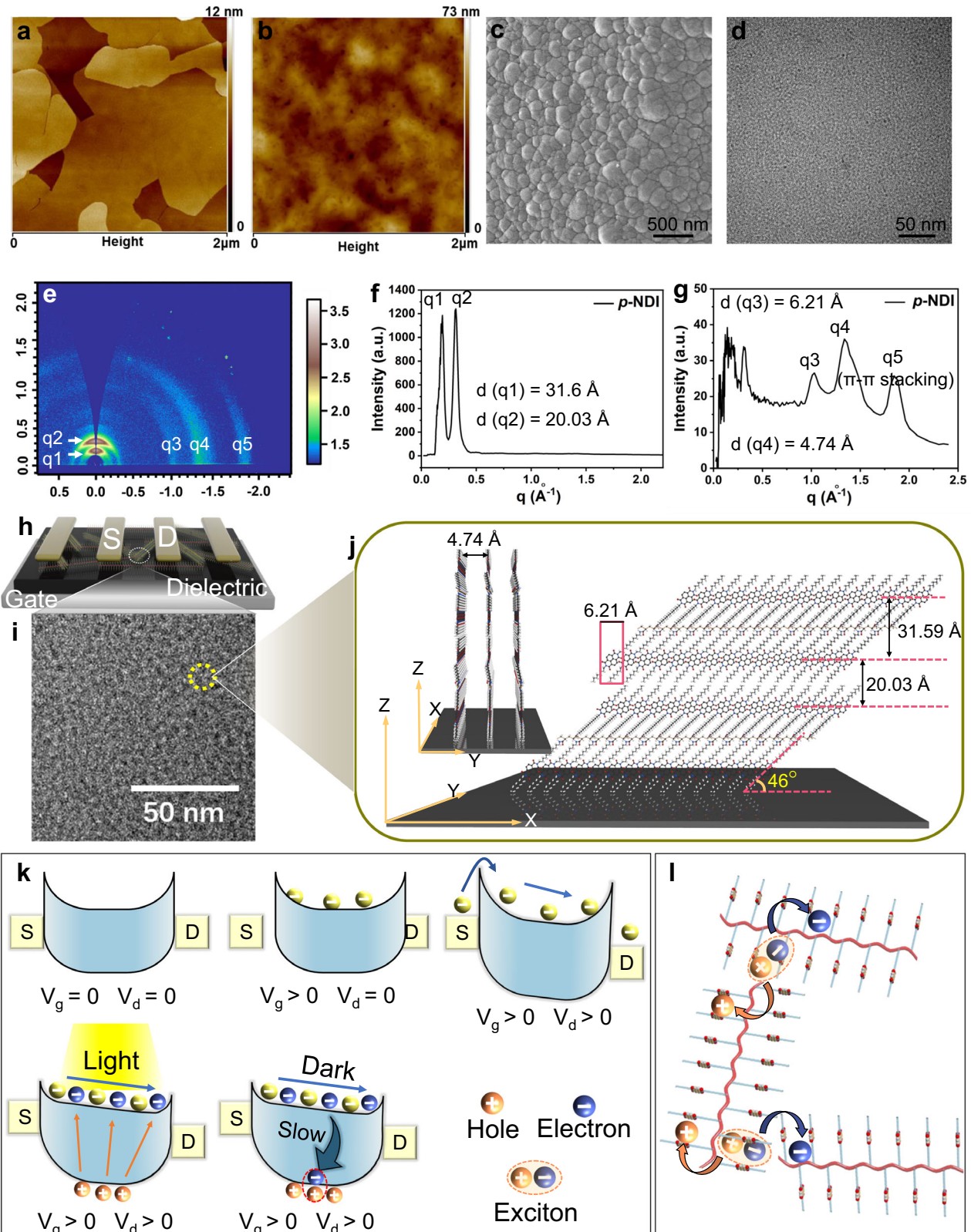

**Fig. 3 | Morphological studies of C8-NDI and *p*-NDI films.** Atomic Force Microscopy (AFM) characterizations of **a** C8-NDI and **b** *p*-NDI films. Scanning Electron Microscopy (SEM) characterization of **c** C8-NDI and **d** *p*-NDI films. **e** 2D grazing-incidence X-ray diffraction (GIXD) of *p*-NDI films, with the out-of-plane diffraction peaks shown in **f**, and in-plane diffraction peaks shown in **g**. **h** The device architecture of *p*-NDI phototransistor. **i** The enlarged view of **d**. **j** The molecular packing, ordering, and orientation of *p*-NDI domains revealed by GIXD characterization. **k** The proposed charge generation and recombination diagram, and the relationship between energy bands and voltages applied on the gate and drain electrodes. **l** The in-plane disordering induced exciton dissociations at the interfaces of polymers in the *p*-NDI film.

generated. Upon switching off the light, $I_d$ decreases rapidly due to the recombination of the electrons with holes in shallow traps, which characterizes the early stage of the relaxation. Subsequently, the decay process becomes limited by the poor recombination of electrons with holes in deeper traps (Fig. 3k). Compared with the conventional semiconductors, $p$-NDI has more traps due to its TSCT mode and in-plane disordered morphology, and thus has a greater chance of charge trapping. The holes trapped in the deep-level trapping states can be slowly released and recombined after switching off the light (Supplementary Figs. 18–20). It has been reported that those trapped holes can lead to the space charge gating effect and back gate screen effect[63], and therefore lead to a fading memory. As a result, the synergistic effect of in-plane disordering and out-of-plane ordering renders a high exciton dissociation rate, a reasonable mobility, and a low charge recombination rate. All these properties contribute to the unique photo responses and fading memory of $p$-NDI.

Time-dependent fluorescent spectrum of $p$-NDI film during light irradiation provides further evidence for its efficient exciton separation. As shown in Supplementary Fig. 21, the fluorescent emission peak intensity of $p$-NDI film gradually increases to 188% of the initial value after 4 min of light irradiation. However, C8-NDI film shows a 12% peak intensity drop in its emission spectrum at identical conditions. Note that both $p$-NDI and C8-NDI are chemically stable at given experimental conditions. It is reported that the exciton quenching in the aggregated states would decrease the emission intensity[51]. Hence, it is reasonable to conclude that efficient exciton dissociations occur in the $p$-NDI films, whereas exciton quenching dominates the photophysical process in the C8-NDI films. Moreover, the time-resolved photoluminescence (TR-PL) analysis of the $p$-NDI film and three other conventional semiconductor films offers more insights on the fading memory. The results show a τ (the intensity weighted average carrier lifetime) of 0.708, 0.979, 1.121, and 1.399 ns for C8-NDI, pentacene, P3HT, and $p$-NDI, respectively (Supplementary Fig. 22). The largest τ value of $p$-NDI among all semiconductors affirms the photogenerated excitons in $p$-NDI are easier transferred and less recombined than three other conventional semiconductors[41], thus leading to better-separated photocurrent responses and a fading memory.

## Signal sensing, memorizing, and pre-processing by $p$-NDI transistor

In contrast to conventional photodetectors, $p$-NDI phototransistors exhibit light dosage-dependent output currents and retention time, therefore enabling multiple complicated image pre-processing functions, such as image integration, weak signal accumulation, and spectrum analysis[3]. To assess the retina-like signal pre-processing capability of $p$-NDI transistor, we apply a light pulse with an intensity of 4200 µW cm$^{-2}$ to a $p$-NDI transistor array through a photomask with a "F" pattern. Here, one $p$-NDI transistor corresponds to one image pixel. As shown in Fig. 4a, the irradiated pixels gain higher $I_{ds}$ than that of the masked pixels. The longer the light pulse, the higher $I_{ds}$ of the irradiated pixels. As a result, the letter "F" is sensed and memorized in the array after removing the light pulse.

Additionally, this array shows a contrast enhancement function, being essential to highlight the key feature of an image[3]. In Fig. 4b, an image with four gray scales is input into a 3 × 3 $p$-NDI transistor array by applying four different light intensities to the given pixels, i.e., 2800 µW cm$^{-2}$ (Pixel A), 700 µW cm$^{-2}$ (Pixel B), 70 µW cm$^{-2}$ (Pixel C), and 0 µW cm$^{-2}$ (Pixel D). The normalized $I_{ds}$ of pixel A, B, C, and D is 1.00, 0.83, 0.35, and 0, respectively. As the memory fades, the normalized $I_{ds}$ of pixels A, B, C, and D change to 1.00, 0.81, 0.30, and 0 at 1 min, 1.00, 0.53, 0, and 0 at 10 min, and 1, 0, 0, and 0 at 20 min, respectively. The above results indicate that the device irradiated with a higher light intensity gains a higher $I_{ds}$ and thus exhibits a longer retention time. The $I_{ds}$ corresponding to pixels with a lower intensity decay faster, whereas the $I_{ds}$ from pixels with a higher intensity decay slower. The $I_{ds}$

differences between four pixels are enlarged over time, hence leading to an image with an enhanced contrast and highlighted key features.

## $p$-NDI transistor for optoelectronic in-sensor RC

The nonlinear dynamic evolution of channel current upon the light stimulus of $p$-NDI is a key requirement of the in-sensor RC. Such dynamic evolution, characterized by the short-term memory, is revealed by applying different optical pulse trains to the transistor (Fig. 4c and Supplementary Fig. 23). Each optical pulse train contains five pulses, where '1' ('0') denotes an optical pulse with a light intensity of 70 µW cm$^{-2}$ (0 µW cm$^{-2}$) and a pulse width of 0.5 s. Upon receiving a pulse '1', an increase in current is detected, whereas a pulse '0' yields a current decrease. Attributing to the efficient exciton dissociation at the interface of $p$-NDI domains with different orientations, the $I_{ds}$ increases continuously over consecutive light pulses, rather than a fast saturation. After removing light irradiation, the free holes and electrons recombine slowly, resulting in a nonlinear decay of output $I_{ds}$ and thus a short-term memory. Therefore, the combined effect of photogating and fading memory enables current facilitation/depression and dynamic transistor states with well-separated outputs. As demonstrated in Fig. 4c, thirty-two optical pulse trains ranging from (00000) to (11111) generate 32 clearly distinguishable states, implying a robust capability of mapping the complicated spatiotemporal signals into reservoir states, which is crucial for multi-task learning[25,36]. Additionally, the optoelectrical RC is flexible and wearable, as shown in Fig. 4d, indicating the unique advantages of the $p$-NDI semiconductor.

## Multi-task in-sensor computing

The optoelectrical in-sensor RC for multi-task computing is schematically illustrated in Fig. 5a. Optical stimuli representing different images/tasks is directly input into the RC without any analog-to-digital data conversion. Each neuron in the reservoir (ruby nodes in Fig. 5f) is a delayed feedback dynamic node physically implemented on a $p$-NDI transistor, which requires no training. Only the lightweight connections of readout map (colored straight arrows) linking the reservoir neurons and output neurons (right green/yellow/blue nodes pointed by the colored arrows in Fig. 5f) are optimized during learning. These features thereby greatly reduce the data processing cost, faithfully resemble the photoreceptor of the human retina, and manifest the unique advantage of optoelectrical RC over conventional neuron networks.

Extensive tests were carried out on the multi-task optoelectronic RC system leveraging both the fading optical memory of $p$-NDI and memristive organic diodes. Figure 5a illustrates the schematic of the multi-task optoelectronic RC system, which shares a strong resemblance with the human visual system in terms of the in-sensor architecture as well as the multi-task capability (e.g., to simultaneously learn different attributes, such as the type and size, of a variety of garments). The optical images of garments are first fed to the organic optoelectronic reservoir to produce semantic features using the intrinsic fading memory, which are the inputs to the organic diode readout map to classify the category of the garments (Supplementary Fig. 24. The synthesis and basic characterizations of the ionic diode materials polyAT and polyES are shown in Supplementary Scheme 3, 4 and Supplementary Fig. 25). The same reservoir and readout maps are then used to identify the size of the garments, via alphanumeric letters such as 'S', 'M', and 'L' for clothes and digits for shoes. The circuit of the $p$-NDI reservoir is schematically illustrated in Fig. 5b. Different light signals are received by $p$-NDI transistors to generate different source-drain currents, which are converted to voltages by trans-impedance amplifiers before being digitized by analog-digital convertors (ADC). The post-processed digital signals are then fed to the digital-analog convertors (DACs) that drive the memristive diode crossbar array to implement the readout layer as shown in Fig. 5c. Each cell of the array consists of a single memristive diode, acting as the tunable weight of the readout layer. Figure 5d shows the optical garment images

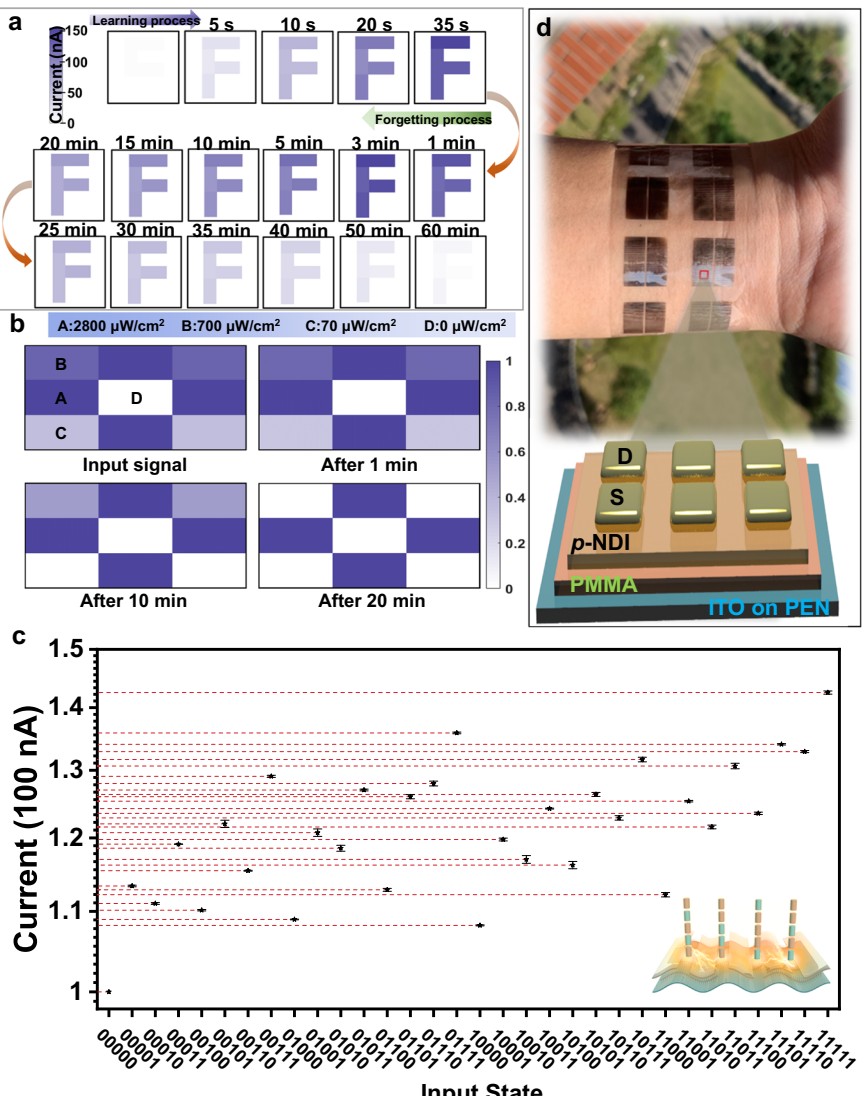

**Fig. 4 | Signal sensing, memorizing, and pre-processing by *p*-NDI transistor.** **a** Illustrations of the image learning and forgetting of the letter 'F'. The letter 'F' was stimulated with a light intensity of 4200 μW cm⁻². **b** An illustration of the image contrast enhancement after the training processes. **c** Experimental read-current responses generated by thirty-two 5-bit optical pulse trains ranging from (00000) to (11111). The insert indicates the light pulse train and flexible *p*-NDI transistor. Each data point is the average of 12 repeated tests. All statistical data are presented as a mean ± SD (*n* = 12). **d** Image of wearable RC based on *p*-NDI phototransistor with its architecture schematically illustrated.

sampled from the representative Fashion-MNIST dataset[64], while size information is based on the letter images from the E-MNIST letters dataset[65] and digit images from the MNIST dataset (Supplementary Fig. 26)[66]. For simplicity, all optical images possess 28 pixels both vertically and horizontally. As shown in Fig. 5e, these images are first binarized and trimmed to remove edge pixels which are mostly blank. The remaining pixels are reshaped into 140-wide 5-long optical sequences and fed to the dynamic reservoir of 140 visual neurons made of *p*-NDI transistors (see Fig. 5f). The optoelectronic reservoir first perceives the optical input, acting as the retina, and then extracts visual features of the input signals, playing a similar role to the primitive visual cortex or visual area 1. The state of the reservoir evolves according to the fading memory dynamics of the *p*-NDI transistors, which transforms the temporal sequences of the input optical signals into a 140-long visual feature in the form of electric currents as shown in Fig. 5f. This visual feature is then received by the organic readout map, acting like the striate cortex, for recognition. The multi-task readout map first classifies the category of the garment with the garment-specific readout classifier, thanks to the non-volatile tunable

characteristic of organic diode memristors playing the role of a fully connected detection head (Supplementary Fig. 24). Once the system recognizes upper garments or pants, the letter-specific organic classifier system will be activated and fed with the corresponding visual feature. Otherwise, if shoes are recognized, the digit-specific organic classifier will be wakened up to process the corresponding visual feature (see Supplementary Fig. 27 for the readout maps of these three tasks). Figure 5g reports the recognition rate, which is 88.00% for the system to simultaneously recognize not only the garments but also their sizes (The independent accuracies are 98.04%, 88.18%, and 91.76% for recognizing the letters[35], the digits[8,34,36], and the garments, respectively, as shown in Fig. 6e), which is only slightly lower than that of the pure software reservoir-computing counterpart (98.71% on recognizing letters, 91.07% on numbers, 92.08% on garments, and 89.38% on overall multi-tasking classification. See methods for details of the software baseline). The accuracies for most classes are more than 80% as evidenced by the chart in Fig. 5g. The performance is retained even under bending condition, where the *p*-NDI transistor-based in-sensor RC could still well discriminate 32 digits

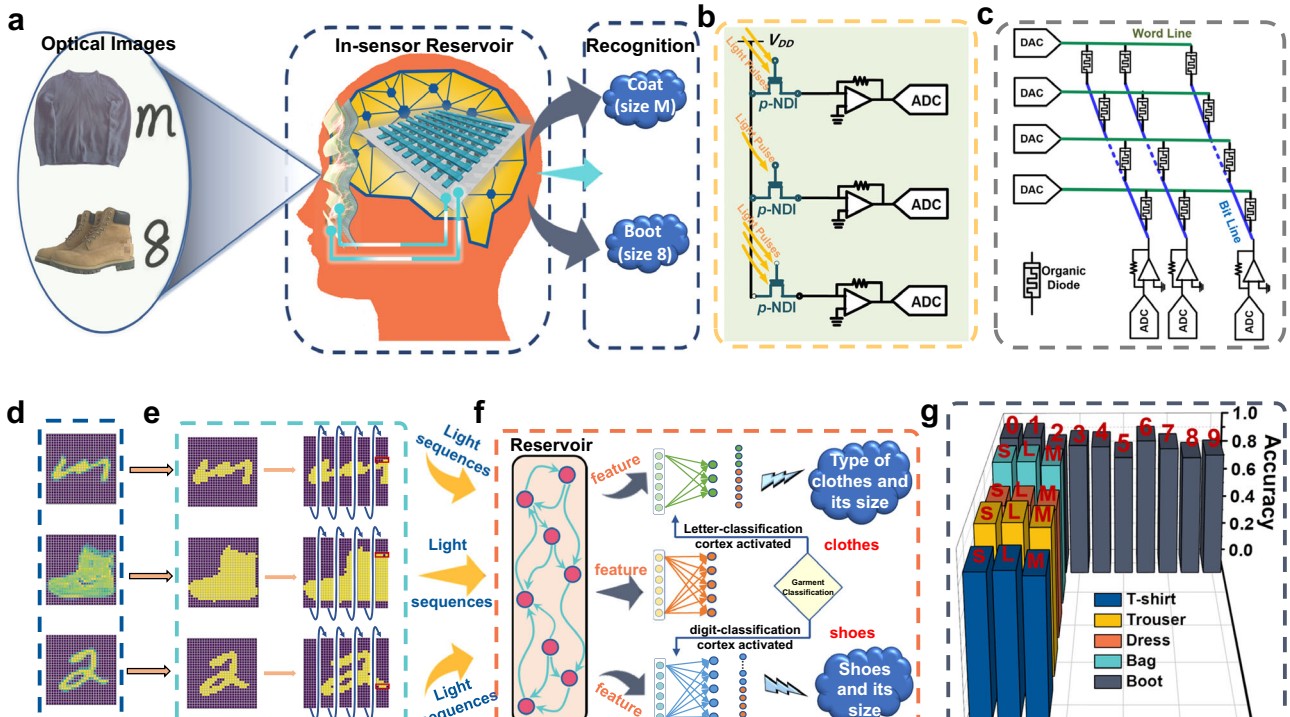

**Fig. 5 | The multi-task reservoir-computing framework. a** Illustration of optical multi-task recognition. The optical garment images with associated size information are inputs to the *p*-NDI optoelectronic reservoir that are classified by memristive organic diode-based readout maps. **b** The *p*-NDI circuits performing as reservoir. **c** The circuit of ionic diodes crossbar performing the readout layer. **d** Optical images of garments and size labels, sampled from the representative F-MNIST, E-MNIST, and MNIST datasets. **e** Software pre-processing of the input images to be fed to the organic optical reservoir. **f** The *p*-NDI optical reservoir, which extracts discriminative features for all three sub-tasks without training thanks to the fading memory, the task-wise memristive organic diode readout maps, and the multi-task classification flow. **g** The multi-task recognition result. The system achieved a performance comparable to the software baseline.

(Supplementary Fig. 28) and give an overall accuracy of 83.67% (The independent accuracies are 97.25%, 85.37%, and 88.48% for recognizing the letters, the digits, and the garments, respectively, as shown in Supplementary Table 1.), close to that of the device without experiencing any mechanical deformation. Comparatively, for the P3HT (and many other conventional semiconducting polymers) transistor with a bare $SiO_2$ dielectric layer, though it shows a fading memory that stemmed from the surficial traps on the $SiO_2$, it is still very difficult to clearly separate 32 digits due to a poor separation in $I_d$ (Supplementary Fig. 29). As a result, P3HT/$SiO_2$ transistor-based in-sensor RC exhibits an inferior performance in multi-task learning and recognition. For example, the accuracies of recognizing handwritten letters and numbers, and classifying a variety of costumes is 95.75%, 81.75%, and 86.08%, respectively. The overall accuracy is 77.86% for the system to correctly recognize not only the garments but also their size (Supplementary Fig. 30 and Supplementary Table 1). All these values are much lower than that of *p*-NDI transistor-based RC system.

We further benchmark the performance and efficiency of the multi-task RC system. Figure 6a shows the feature vectors of all the 50 thousand test samples extracted by the *p*-NDI optoelectronic reservoir. Each of the 280-long columns represents a feature vector of the associated garment and size, which is grouped according to the labeled class. It is clear that the feature vectors from of the same class are similar while the feature vectors of different classes are dislike, illustrating the discriminative power of the optoelectronic reservoir. Figure 6b shows the zoomed details of feature vectors from the class of small-size T-shirts and large-size trousers, where a clear border separates the features from these two different classes. To better visualize the distribution of these high-dimensional feature vectors encoded by the reservoir, they are reduced to points in three-dimensional spaces via linear discriminant analysis (LDA), where the point color denotes

the corresponding class of the garment and its size, as shown in Fig. 6c and Supplementary Fig. 31. The samples from the same class are typically clustered together while those from the different classes are generally separated. Figure 6d shows the accuracy and weight efficiency of our system compared with the single-layered artificial neural network (ANN) and the double-layered ANN. The overall performance of our optoelectronic reservoir-computing system achieves 88.00%, which is comparable to the single-layered ANN (92.40%), and double-layered ANN (94.51%). Thanks to the affordable training of RC, the total number of weights in our system is ~2.6k, while that of the single-layered ANN is ~12.6k and that of the double-layered ANN is ~105.9k. We then measure the weight efficiency of our system by computing the accuracy per weight, which is 0.033, 0.0073, and 0.00089 for our system, single-layered ANN, and double-layered ANN, respectively, implying our system achieves the best performance using the same population of network parameters. Figure 6e shows the independent confusion matrices. The independent accuracies are 88.18%, 98.04%, and 91.76% for the MNIST, EMNIST-letters, and FMNIST datasets, respectively. All the matrix rows are dominated by their diagonal elements, implying the high class-wise accuracy on each sub-task (see Supplementary Fig. 32 for non-normalized confusion matrices, and Supplementary Fig. 33 for the overall confusion matrix of the multi-task learning). Figure 6f compares the number of multiplication and accumulation (MAC) operations during the training and inference of our optoelectronic reservoir-computing system, the single-layered ANN, and double-layered ANN. During the inference, the number of MAC operations for a single optical image is ~2.9k, which is significantly less than that of the single-layered ANN (~14.4k) and the double-layered ANN (~105.9k). Similarly, in the training, the number of MAC operations of our system is ~6.7k, compared to ~33.3k of the single-layered ANN and ~265.2k of the double-layered ANN.

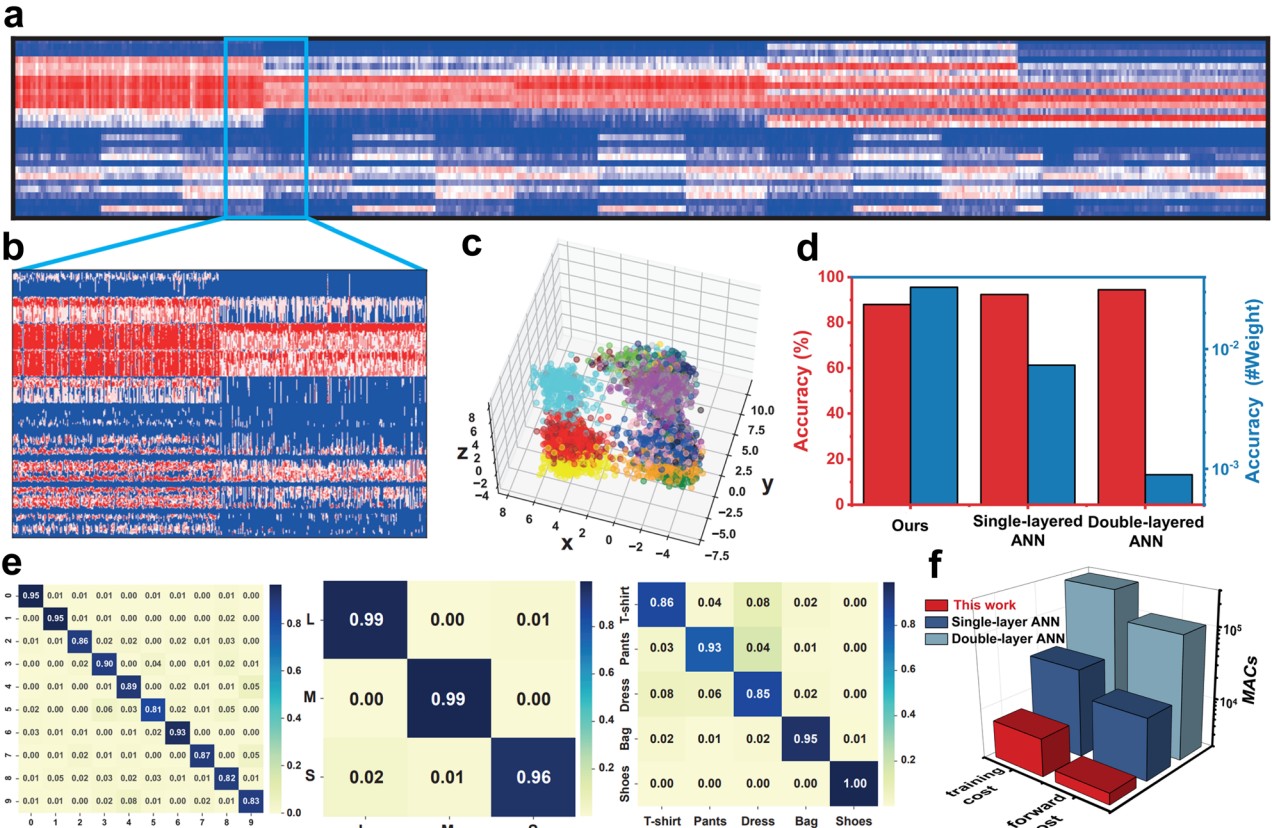

**Fig. 6 | Multi-task classification performance. a** The stacked feature vectors of each class (a combination of sub-tasks). Each column represents the feature of a sample retrieved by the organic optical reservoir. **b** The details of the features of two different classes. **c** The 3D visualization of the feature vector distribution using linear discriminative analysis (LDA) for dimensionality reduction. **d** Comparison of the accuracy and number of trainable weights between our optoelectronic reservoir-computing system, a single-layered ANN, and a double-layered ANN. Our system achieves comparable performance with drastically reduced number of trainable parameters. **e** Confusion matrices of the reservoir-computing system on the three sub-tasks. The system achieves sound accuracy in all sub-tasks where matrix rows are dominated by the diagonal elements. **f** Comparison of training and inference costs among our optoelectronic RC system, a single-layered ANN, and a double-layered ANN. Our optoelectronic RC system features the least MAC operations in both training and inference.

In addition to 2D images, the unique spatiotemporal dynamics of wearable in-sensor RC could be used to extract features of videos, such as those captured by dynamic vision sensors (DVS). Here we focus on three different hand gestures (hand clapping, right-hand waving, and left-hand waving), a subset of the widely used DVS128 Gesture dataset[67]. An example event stream about the left-hand waving is shown in Fig. 7a (the spatial view of accumulated events) and Fig. 7b (the spatiotemporal view) (see Supplementary Video). Each point in the figures represents an event triggered by the illumination intensity change of the associated DVS pixel, as shown in Fig. 7c. The optical event streams are physically processed by a proposed 28 × 28 $p$-NDI transistor array, functioning not only as an image sensor but also a reservoir computer, as shown in Fig. 7d. Figure 7e is the extracted 2D feature of the 3D event stream by the $p$-NDI in-sensor RC system, equivalent to video compression. This 28 × 28 feature map is fed to a linear classifier, the non-volatile organic readout layer, as shown in Fig. 7f (Common interconnects between $p$-NDI array and organic memristor array are remarked by integers). During training, the read-out layer is optimized via pseudo-inverse, which suits real-time edge learning. For inference, the optimized readout map takes a reservoir feature map and outputs a classification vector, in the form of analog currents that are subsequently digitized in Fig. 7f, g, predicting the gesture type behind the event stream (left-hand waving in this case). In this case, the event video sample in Fig. 7a is predicted to be left-hand waving with a 100% confidence. To demonstrate the advantage of the reservoir in video feature extraction, we visualize the distribution of reservoir feature maps of different hand gestures via PCA, as shown in

Fig. 7h. The samples from the same class are close to each other, while the samples from different classes are well separated. The overall accuracy is 98.62% for the system to correctly recognize the three gestures. The detailed class-wise accuracy and error rate are reported in Fig. 7i, where diagonal elements dominate each row, indicating a high accuracy in each gesture classification.

## Discussion

Attributing to the unique bottle-brush configuration and through-space charge-transport characteristics of $p$-NDI, as well as the good out-of-plane orientation and poor in-plane ordering of $p$-NDI semi-conducting film, efficient exciton dissociations and slow charge recombination have been realized. Relying on these features, the $p$-NDI phototransistors exhibit excellent light-responsive behavior and non-linear fading memory, showing the ability to in situ sense, memorize, and simultaneously pre-process the optical inputs. Moreover, their well-separated light responses, fading memory, and echo state property enable a wearable transistor-based dynamic RC system. Paired with a 'readout function' implemented on memristive organic ionic diodes, the all-organic-materials-based optoelectrical RC achieves accuracies of 98.04%, 88.18%, and 91.76% in recognizing handwritten letters and digits, and classifying a variety of costumes, which translates to an accuracy of 88.00% in simultaneously recognizing different attributes of garments (among the highest value for organic semi-conductors, only slightly lower than the accuracy 89.38% of its pure software counterpart), highlighting its robust multi-task learning capability. Further, the training cost of this $p$-NDI-based RC is

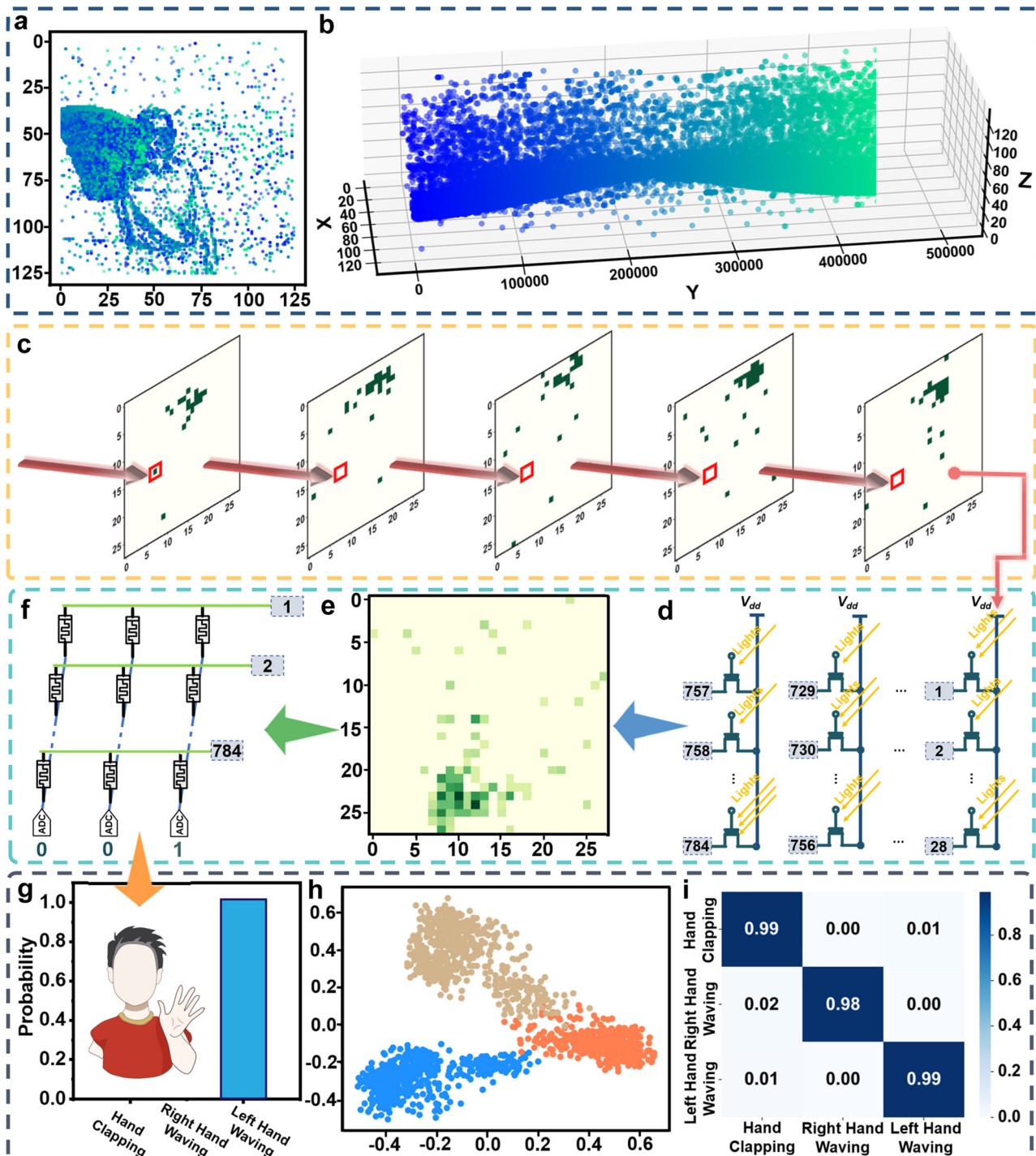

**Fig. 7 | Event-based video classification with DVSGesture128 dataset.**
A sample of the left-hand waving event stream is shown in **a** spatial view and
**b** spatiotemporal view. The event video is down-sampled into 28 pixels in both
spatial directions and time-binned into 32 frames. Five evenly spaced frames **c** are
perceived by the in-sensor RC system **d**. **e** The feature map extracted from the in-
sensor reservoir matrix. **f** The non-volatile organic memristor array, functioning as
the readout map. **g** The classification results, a [0, 0, 1] vector in this case, repre-
senting the probabilities that the sample falls to the 3 classes. **h** The feature dis-
tribution visualization via PCA showing the feature map of different gestures
extracted from the reservoir system can be linearly divided even in the two-
dimensional space. Each point represents a sample and the color represents its
class. **i** The confusion matrix, dominated by the diagonal elements.

significantly lower than that of ANN counterparts, making it suitable
for edge scenarios. Moreover, the RC is able to efficiently classify the
left-hand waving, right-hand waving, and hand clapping gestures with
an overall accuracy of 98.62%. This work not only overcomes the
bottleneck associated with conventional sensing-computing systems
of large time and energy overheads, but also provides a promising
material-algorithm co-design strategy for wearable, affordable, and

highly efficient photonic neuromorphic systems with multi-task
learning capability.

## Methods
The instruments and equipment for all characterizations, the device
fabrication and characterization, and simulations of the reservoir
computing are detailed in the Supplementary Information.

## Data availability

The authors declare that all relevant data are available in this paper and its Supplementary Materials and that data supporting the results of this study are available from corresponding authors upon reasonable request. Source data are provided with this paper.

## Code availability

Code for *p*-NDI-based reservoir computing for multi-task learning and DVS gesture recognition is available at https://github.com/wangsc1912/rc_pndi.

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

## Acknowledgements

This work was supported by National Natural Science Foundation of China (22275193 (W.G.H.)), the Natural Science Foundation of Fujian Province (E131AJ0101 (W.G.H.)), Fujian Science and Technology Innovation Laboratory for Optoelectronic Information of China (2021ZR115 (W.G.H.)), Fujian Institute of Research on the Structure of Matter, Chinese Academy of Sciences (E055AJ01 (W.G.H.)), National Natural Science Foundation of China (62122004 (Z.W.)), Hong Kong Research Grant Council (27206321, 17205922 (Z.W.)), and Beijing Natural Science Foundation (Z210006 (Z.W.)). This research is also partially supported by ACCESS—AI Chip Center for Emerging Smart Systems, sponsored by Innovation and Technology Fund (ITF), Hong Kong SAR.

## Author contributions

X.S.W. and S.C.W. made equal contributions to this work. X.S.W. and S.C.W. conducted the experiments. W.H. performed XRD experiments and data visualization. Y.D. synthesized ionic gels, W.G.H. and Z.R.W. initiated the idea and wrote the manuscript. All authors contributed to the data analyses.

## Competing interests

The authors declare no competing interests.
