## [Peer Review File · Nature Communications]

REVIEWER COMMENTS

Reviewer #1 (Remarks to the Author):

The paper titled "Wearable in-sensor reservoir computing using optoelectronic polymers with through-space charge-transport characteristics for multi-task learning" is well thought out, well put together, and in accordance with current novel optoelectronic device fabrication and implementation techniques.

Paper manuscript requires minor revisions in regard to grammar and spelling errors.

Please elaborate on how you thermally deposited C8-NDI, C8-NDI-C12 and pentacene in supplemental. Why did you thermal evaporate these organic molecules but spin coat p-NDI and P3HT? If C8-NDI and C8-NDI-C8 are synonymous, please correct to be same and match through paper and supplemental (Figure S1 g-h).

What is thickness of all p-NDI semiconducting layer? You list all others in supplemental.

Was fabrication of devices performed in an inert atmosphere or at ambient conditions?

HOMO calculations do not match up with text. i.e. $E_{\text{HOMO}} = h\nu - (E_{\text{cutoff}} - E_{\text{onset}})$. You list E_{cutoff} and then refer to that value as your HOMO. Please elaborate why or correct.

In manuscript on page 14, line 352 and 355, in reference to Figure 5f colors do not match. Please correct.

Reviewer #2 (Remarks to the Author):

Wu et al presented an optical sensor with reservoir computing. By utilizing the non-linear function of optical sensor output, "physical" reservoir computing was successfully demonstrated. The

manuscript is well organized, and the results/discussion are comprehensive. In particular, organic phototransistor with high memory effect after stopping light irradiation was used to realize high accuracy object recognition. Although there is one critical question about the continuous use described below, the reviewer recommends this for publication in this journal if the critical question is solved.

1. For the reservoir computing analyses, long tail of optical sensor output after stopping light irradiation was used. Depending on the light intensity, this long tail length is different, and it takes more than hundreds of second. In this case, if the multiple flashing lights are irradiated with a frequency >1 Hz, what is the happened for the RC analyses. This is very important for the optical imager. Without solving this, it is hard to use this method.

2. The sensor has a mechanical flexibility. Under bending condition, is there any change of sensor response? Furthermore, under bending condition, how good is accuracy to recognize the images?

3. Since this is wearable sensor application, at least one demonstration to recognize the image movement or so in real-time (maybe movie) should be conducted.

Dear Reviewers,

First of all, thank you very much for the constructive comments and helpful suggestions. To address technical concerns, we have conducted new experiments, which are reflected in the revised manuscript and supplementary materials. The point-by-point responses to the comments are as follows.

REVIEWER COMMENTS

Reviewer #1 (Remarks to the Author):

The paper titled "Wearable in-sensor reservoir computing using optoelectronic polymers with through-space charge-transport characteristics for multi-task learning" is well thought out, well put together, and in accordance with current novel optoelectronic device fabrication and implementation techniques.

Answer: We appreciate the positive comments on the significance of our work.

1. Paper manuscript requires minor revisions in regard to grammar and spelling errors.

Answer: We thank the reviewer for this suggestion. The manuscript and supplementary materials have been proofread for grammar and spelling errors, which are highlighted in yellow in the revised version. Some of the changes are as follow.

Page	Line	Before revision	After revision
2	44	analogue	analog
3	78	enables	enable
4	107	feature for	feature
5	122	surprisedly	surprisingly
6	144	different with	different from
11	258	used	using
11	260	giving	given
12	289	release	released
15	340	pixel	pixels
19	451	classifies	classifying

2. Please elaborate on how you thermally deposited C8-NDI, C8-NDI-C12 and pentacene in supplemental. Why did you thermal evaporate these organic molecules but spin coat p-NDI and P3HT? If C8-NDI and C8-NDI-C8 are synonymous, please correct to be same and match through paper and supplemental (Supplementary Fig. 1 g-h).

Answer: We thank the reviewer for the suggestion, which improves the clarity of the manuscript. The details of thermal evaporation are as follows. The evaporation chamber was of a pressure lower than 5×10^{-6} Torr. The semiconductor solid (triple sublimed) was thermally evaporated directly onto a substrate with a thickness of 50 nm at a rate of 0.5 \AA s^{-1} , where the substrate temperature during the deposition was held constantly at 30°C. We have incorporated the details about thermal deposition of C8-NDI, C8-NDI-C12, and pentacene into the revised supplementary materials and highlighted the details in yellow.

Regarding the deposition technique difference, C8-NDI, C8-NDI-C12, and pentacene are small molecules which can be thermally evaporated in high vacuum. However, *p*-NDI and P3HT are polymers with molecular weight over 10 kDa, and therefore cannot be thermally evaporated in high vacuum. Hence, we thermally evaporated C8-NDI, C8-NDI-C12, and pentacene but spin-coated *p*-NDI and P3HT.

As correctly pointed out by the reviewer, here C8-NDI and C8-NDI-C8 refer to the same compound. We have replaced C8-NDI-C8 with C8-NDI throughout the manuscript and revised Supplementary Fig. 1g-h in the supplementary materials.

3. What is thickness of all *p*-NDI semiconducting layer? You list all others in supplemental.

Answer: The thickness of all *p*-NDI semiconducting layer is 32 ± 2 nm (data from five tests by a Bruker DEKTAK XT profile meter). We have included this information into revised supplementary materials on page 3.

4. Was fabrication of devices performed in an inert atmosphere or at ambient conditions?

Answer: The device fabrications were performed in a glove box filled with N₂, rather than in ambient condition. We have included this information into the corresponding paragraphs of the revised supplementary materials.

5. HOMO calculations do not match up with text. i.e. $E_{\text{HOMO}} = h\nu - (E_{\text{cutoff}} - E_{\text{onset}})$. You list E_{cutoff} and then refer to that value as your HOMO. Please elaborate why or correct.

Answer: Thank you very much for pointing out this error. The E_{cutoff} of C8-NDI and *p*-NDI are 16.65 eV and 16.43 eV, respectively. We have corrected all these numerical values in the revised supplementary materials on page 7.

6. In manuscript on page 14, line 352 and 355, in reference to Figure 5f colors do not match. Please correct.

Answer: Thank you for pointing out this inconsistency. As advised, we have recolored the nodes in the reservoir to better convey the idea in Figure 5f (also shown in Figure R1), and revised the sentence citing the Figure 5f as “Each neuron in the reservoir (ruby nodes in Figure 5f) is a delayed feedback dynamic node physically implemented on a *p*-NDI transistor, which requires no training. Only the lightweight connections of readout map (colored straight arrows) linking the reservoir neurons and output neurons (right green/yellow/blue nodes pointed by the colored arrows in Figure 5f) are optimized during learning.” in the revised manuscript.

Figure R1. The p -NDI optical reservoir, which extracts discriminative features for all three sub-tasks without training thanks to the fading memory, the task-wise memristive organic diode readout maps, and the multi-task classification flow.

Reviewer #2 (Remarks to the Author):

Wu et al presented an optical sensor with reservoir computing. By utilizing the non-linear function of optical sensor output, “physical” reservoir computing was successfully demonstrated. The manuscript is well organized, and the results/discussion are comprehensive. In particular, organic phototransistor with high memory effect after stopping light irradiation was used to realize high accuracy object recognition. Although there is one critical question about the continuous use described below, the reviewer recommends this for publication in this journal if the critical question is solved.

1. For the reservoir computing analyses, long tail of optical sensor output after stopping light irradiation was used. Depending on the light intensity, this long tail length is different, and it takes more than hundreds of second. In this case, if the multiple flashing lights are irradiated with a frequency >1 Hz, what is the happened for the RC analyses. This is very important for the optical imager. Without solving this, it is hard to use this method.

Answer: We thank the reviewer for this question. Our in-sensor reservoir can retain the performance in processing optical stimulus (*e.g.*, image frames) at a higher frequency. In addition, the p -NDI transistor features photo responses that can be tailored by semiconductor engineering. Moreover, the transistors can be electrically reset to re-instate photocurrent between light pulse trains, as detailed in the follow.

1) *Invariant performance of p -NDI transistor-based in-sensor RC with light pulses at a frequency of 2 Hz*

Similar to the experiments described in the manuscript, each optical train contains five optical pulses. The optical pulses are applied to the transistor every 0.5 s, corresponding to a frequency

of 2 Hz. Here, '1' denotes an optical pulse with a light intensity of $70 \mu\text{W cm}^{-2}$, while '0' denotes an optical pulse with a light intensity of $0 \mu\text{W cm}^{-2}$. Upon receiving a pulse '1', there will be a current increase. Whereas a pulse '0' yields a current decrease. As shown in Figure R2, thirty-two optical pulse train inputs (with a frequency of 2 Hz) ranging from (00000) to (11111) generates thirty-two clearly distinguishable states, implying a noise robust mapping of the complicated spatiotemporal signals to the physical reservoir states. Based on the experimentally observed reservoir states, we applied the RC to recognize letters, numbers, and garments from the same database, but at an elevated optical pulsing frequency of 2 Hz. The resulting independent accuracies are 98.04%, 88.18%, and 91.76% for recognizing the letters, the digits, and the garments, respectively. The overall accuracy is 88.00% for the system to correctly recognize not only the type of garments but also their sizes. This is nearly the same with the performance when optical pulses are applied every 2 s (*i.e.*, 0.5 Hz in Table R1), implying that the RC could well maintain the accuracy even if the multiple flashing lights are irradiated with a frequency >1 Hz.

We have updated the corresponding discussion on page 14 to 21 of the manuscript with the RC performance at 2 Hz optical pulsing, and revised Fig. 4c, 5g, 6, and Supplementary Fig. 23, 27, 31, 32, and 33 accordingly.

2) *Engineering the semiconducting layer to facilitate tune the photo response and the photocurrent decay rate of the memory*

The photo response and photocurrent decay rate of *p*-NDI transistor can be modulated by semiconductor engineering to better accommodate high-frequency optical pulses. As mentioned in the manuscript, the through-space charge-transport (TSCT) characteristics of *p*-NDI semiconducting layer leads to slow recombination of charge carriers and thus a fading memory. In sharp contrast, the high charge recombination rate in the highly crystalline C8-NDI semiconducting layer yields a fast photocurrent quenching without fading memory. Leveraging the complementary behaviors *p*-NDI and C8-NDI, the photocurrent decay rate could be readily adjusted by blending C8-NDI and *p*-NDI of the semiconducting layer at different ratios. To this end, we fabricated a series of photo transistors with blended C8-NDI/*p*-NDI semiconducting layer at different C8-NDI weight percentage of 2.5%, 5%, 10%, 25%, 70%, and 90%, followed by characterizing their photo response and decay behaviors at identical conditions. The results are shown in Figure R3a. As a control, the data for 0% C8-NDI, *i.e.*, solo *p*-NDI semiconducting layer, is also included. The time required to fully recover the I_d to the baseline decreases notably from over 55 s to 37, 26, 17, 8, 5, and 3 s when increases C8-NDI weight percentage from 0% to 2.5%, 5%, 10%, 25%, 70%, and 90%, respectively, implying a wide range tunability of the photocurrent decay rate. Such tunability is beneficial for processing optical pulses of different pulsing frequencies and application scenarios.

3) *Electrically resetting the transistor to zero the residual photocurrent between light pulse trains*

Apart from above-mentioned semiconductor engineering strategy, *p*-NDI transistors could be reset to clear the residual photocurrent of the previous light pulse train. As shown in Figure R3b, the slowly decaying photocurrent after light removal undergoes a rapid recovery upon receiving an electrical pulse ($V_g = -100$ V, 0.1 s). The device after reset is ready for receiving the next optical pulse train and can repeatedly respond to new optical pulses with similar initial photocurrents. Moreover, the timing and frequency to reset can be customized according to the underlying optical pulse frequencies, application scenarios, and reservoir parameters.

To clarify points 2) and 3), we have included corresponding discussions into the revised manuscript on page 9, and added Figure R3 in the revised supplementary materials as Supplementary Fig. 15 on page 14.

Figure R2. (a) Experimental read-current responses generated by thirty-two optical pulse train inputs ranging from (00000) to (11111). The optical pulse is applied to the transistor every 0.5 s. Each data point is the average of twelve repeated tests. All statistical data are presented as a mean \pm SD ($n = 12$). (b) Photo-response to thirty-one different pulse streams of *p*-NDI transistor-based RC. (The pulse stream 00000 is the baseline of the curves)

Figure R3. (a) The photocurrent decay curves of C8-NDI/*p*-NDI blend semiconducting layer-based transistors at respective C8-NDI weight percentage of 0%, 2.5%, 5%, 10%, 25%, 70%, and 90%. (b) The recovery of photocurrent after reset by an electrical pulse ($V_g = -100$ V, 0.1 s).

Table R1. The independent and overall accuracies of *p*-NDI based-RC in recognizing the letters, numbers, and the garments at different optical pulse input frequencies.

Frequency	Letter	Number	Garment	Overall
0.5 Hz	97.58%	90.63%	89.34%	87.67%
0.5 Hz (Bent)	97.25%	85.37%	88.48%	83.67%
2 Hz	98.04%	88.18%	91.76%	88.00%

2. The sensor has a mechanical flexibility. Under bending condition, is there any change of sensor response? Furthermore, under bending condition, how good is accuracy to recognize the images?

Answer: We thank the reviewer for this question. As advised by the reviewer, we measured the photo response of the transistor under bending condition, specifically with a bending radius of 5 mm. As shown in Figure R4a-d, the photocurrent response of *p*-NDI transistor at flat and bending states are rather similar, as evidenced by the comparable photocurrent levels and decay times, implying no obvious change of sensor response.

Next, thirty-two optical pulse trains (with a frequency of 0.5 Hz) ranging from (00000) to (11111) were applied to the bent *p*-NDI transistor where the photocurrent responses were recorded. Again, thirty-two distinguishable states were observed, implying a decent capability to map complicated spatiotemporal signals into reservoir states (Figure R4e). Based on that, we applied the bent *p*-NDI transistor-based RC to recognize letters, numbers, and garments from the same database. The resulting independent accuracies are 97.25%, 85.37%, and 88.48% for recognizing the letters, the digits, and the garments, respectively. The overall accuracy is 83.67% for the system to correctly

recognize not only the types of garments but also their sizes. The accuracy is close to that of the device without experiencing any mechanical deformation (Table R1), corroborating the potential for smart soft electronics.

We have included the discussion above into the revised manuscript on page 19, and Figure R4 and Table R1 into supplementary materials as revised Supplementary Fig. 28 and Table 1.

Figure R4. The photocurrent response of *p*-NDI-based transistor under 1 s light irradiation with different intensities at (a) flat and (b) bending states, respectively. The photocurrent response of *p*-NDI-based transistor under 700 $\mu\text{W cm}^{-2}$ light irradiation with different pulse widths at (c) flat and (d) bending states, respectively. The transistor was working in a ‘sampling’ mode at a constant V_g

and V_d of 100 V with a holding time and interval of 0.01 s and 0.1 s, respectively. (e) Experimental read-current responses generated by thirty-two optical pulse train inputs ranging from (00000) to (11111). The optical pulse is applied to the transistor every 2 s. Each data point is the average of twelve repeated tests. All statistical data are presented as a mean \pm SD ($n = 12$).

3. Since this is wearable sensor application, at least one demonstration to recognize the image movement or so in real-time (maybe movie) should be conducted.

Answer: We thank the reviewer for this constructive suggestion. As advised by the reviewer, we have estimated the performance of our wearable in-sensor RC system in classifying event streams about hand gestures captured by dynamic vision sensors (DVS).

Here we focus on 3 different hand gestures (hand clapping, right hand waving, and left hand waving), a subset of the widely used DVS128 Gesture dataset¹. An example event stream about the left-hand-waving is shown in Figure R5a (the front view) and Figure R5b (the time axis view) (see Supplementary Video). Each point in the figures represents an event triggered by the illumination intensity change of that pixel. The event stream is first down-sampled to 28 pixels in both x and z directions, followed by time binning into 32 frames (*i.e.*, integrating the events in 32 equally spaced time intervals). Five evenly spaced frames from the 32 frames are taken for the downstream classification, as shown in Figure R5c. The optical event streams are physically processed by a proposed 28×28 *p*-NDI transistor array, functioning not only as an image sensor but also a reservoir computer, as shown in Figure R5d. Each *p*-NDI transistor pixel translates the 5-frame optical event pulse stream to a source-drain current change. As thus, the final output of the array shown in Figure R5e are 2D features of the 3D event streams, equivalent to video compression. This 28×28 feature map is then flattened into a 784-long vector, before being fed into the non-volatile organic readout layer, as shown in Figure R5f. (Common interconnects between *p*-NDI array and organic memristor array are remarked by integers.) During training, the readout layer is optimized *via* pseudo-inverse, which suits real-time edge learning. For inference, the optimized readout map takes a reservoir feature map and outputs a classification vector, in the form of analog currents that are subsequently digitized in Figure R5 f-g, predicting the gesture type behind the event stream (left-hand waving in this case). In the case, the gesture sample in Figure R5a is classified into the 3rd class (left-hand waving gesture) with a 100% confidence. To demonstrate the effectiveness of the reservoir, we visualize the distribution of reservoir feature map of different hand gestures *via* PCA, as shown in Figure R5h. The samples from the same class are close to each other, while the samples from different classes are well separated. The overall accuracy is 98.62% for the system to correctly recognize the 3 gestures. The detailed class-wise accuracy and error rate are reported in Figure R5i, where diagonal elements dominate each row, indicating a high accuracy in each gesture classification.

We have included these experimental findings and corresponding discussions into the revised manuscript on page 21-23, and add Figure R5 as Figure 7 in the revised manuscript.

Figure R5. The demonstration on the temporal task based on DVSGesture128 dataset. A sample of the left-hand waving stream is shown in (a) spatial view and (b) time-axis view. The event data is down-sampled into 28 pixels both horizontally and vertically and converted into 32 frames, and five of the frames (c) are perceived by the computing-in-sensor RC matrix (d). (e) The feature map extracted from the in-sensor reservoir matrix. (f) The non-volatile organic memristor array, functioning as the readout map. (g) The classification results. (a 0-0-1 vector in this case, representing the given sample is classified into the third class) (h) The feature distribution visualization *via* PCA showing the feature map of different gestures extracted from the reservoir system can be linearly divided even in the two-dimensional space. Each point represents a sample and the color represents its class. (i) The confusion matrix of the reservoir computing system on the task.

References

1. Amir, A., Taba, B., Berg, D., Melano, T., McKinstry, J., Nolfo, C., Nayak, T., Anderopoulos, A., Garreau, G., Mendoza, M., Kusnitz, J., Debole, M., Esser, S., Delbruck, T., Flickner, M., Modha, D. A Low Power, Fully Event-Based Gesture Recognition System. 2017 IEEE Conference on Computer Vision and Pattern Recognition (CVPR), 2017, pp. 7388-7397.

REVIEWERS' COMMENTS

Reviewer #1 (Remarks to the Author):

The paper titled "Wearable in-sensor reservoir computing using optoelectronic polymers with through-space charge-transport characteristics for multi-task learning" is well thought out, well put together, and in accordance with current novel optoelectronic device fabrication and implementation techniques.

The resubmitted work fixed all concerns with prior manuscript. Publish as is; no need for further revisions.

Reviewer #2 (Remarks to the Author):

The authors addressed all concerns raised by reviewers. The manuscript is ready for publication without further change.

Dear Reviewers:

We are happy to see that two reviewers are satisfied with our revisions.

REVIEWERS' COMMENTS

Reviewer #1 (Remarks to the Author):

The paper titled "Wearable in-sensor reservoir computing using optoelectronic polymers with through-space charge-transport characteristics for multi-task learning" is well thought out, well put together, and in accordance with current novel optoelectronic device fabrication and implementation techniques.

The resubmitted work fixed all concerns with prior manuscript. Publish as is; no need for further revisions.

Answer: We are grateful for your positive comments.

Reviewer #2 (Remarks to the Author):

The authors addressed all concerns raised by reviewers. The manuscript is ready for publication without further change.

Answer: We are grateful for your positive comments.